# Site-Specific Dual-Labeling of a VHH with a Chelator and a Photosensitizer for Nuclear Imaging and Targeted Photodynamic Therapy of EGFR-Positive Tumors

**DOI:** 10.3390/cancers13030428

**Published:** 2021-01-23

**Authors:** Emma Renard, Estel Collado Camps, Coline Canovas, Annemarie Kip, Martin Gotthardt, Mark Rijpkema, Franck Denat, Victor Goncalves, Sanne A. M. van Lith

**Affiliations:** 1Institute de Chimie Moléculaire de l’Université de Bourgogne ICMUB UMR CNRS 6302, Université Bourgogne Franche-Comté, 21000 Dijon, France; Emma_Renard@etu.u-bourgogne.fr (E.R.); coline.canovas@gmail.com (C.C.); franck.denat@u-bourgogne.fr (F.D.); victor.goncalves@u-bourgogne.fr (V.G.); 2Department of Biochemistry, Radboud Institute for Molecular Life Sciences, Radboudumc, 6525 GA Nijmegen, The Netherlands; E.ColladoCamps@radboudumc.nl; 3Department of Medical Imaging, Nuclear Medicine, Radboudumc, 6525 GA Nijmegen, The Netherlands; Annemarie.Kip@radboudumc.nl (A.K.); Martin.Gotthardt@radboudumc.nl (M.G.); Mark.Rijpkema@radboudumc.nl (M.R.)

**Keywords:** variable domain of heavy chain only antibodies (VHH), site-specific conjugation, dual-labeling, nuclear imaging, photodynamic therapy

## Abstract

**Simple Summary:**

Variable domains of heavy chain only antibodies are small proteins that can be used for tumor imaging and therapy upon conjugation of functional groups. As frequently used random conjugation techniques can decrease binding to the target of interest, site-specific conjugation of these functional groups is preferred. Here, we optimized site-specific conjugation of both a chelator for binding of a radiometal and a photosensitizer to epidermal growth factor receptor (EGFR) binding VHH 7D12. We characterized this dual-labeled VHH for nuclear imaging and targeted photodynamic therapy of EGFR-expressing tumors.

**Abstract:**

Variable domains of heavy chain only antibodies (VHHs) are valuable agents for application in tumor theranostics upon conjugation to both a diagnostic probe and a therapeutic compound. Here, we optimized site-specific conjugation of the chelator DTPA and the photosensitizer IRDye700DX to anti-epidermal growth factor receptor (EGFR) VHH 7D12, for applications in nuclear imaging and photodynamic therapy. 7D12 was site-specifically equipped with bimodal probe DTPA-tetrazine-IRDye700DX using the dichlorotetrazine conjugation platform. Binding, internalization and light-induced toxicity of DTPA-IRDye700DX-7D12 were determined using EGFR-overexpressing A431 cells. Finally, ex vivo biodistribution of DTPA-IRDye700DX-7D12 in A431 tumor-bearing mice was performed, and tumor homing was visualized with SPECT and fluorescence imaging. DTPA-IRDye700DX-7D12 was retrieved with a protein recovery of 43%, and a degree of labeling of 0.56. Spectral properties of the IRDye700DX were retained upon conjugation. ^111^In-labeled DTPA-IRDye700DX-7D12 bound specifically to A431 cells, and they were effectively killed upon illumination. DTPA-IRDye700DX-7D12 homed to A431 xenografts in vivo, and this could be visualized with both SPECT and fluorescence imaging. In conclusion, the dichlorotetrazine platform offers a feasible method for site-specific dual-labeling of VHH 7D12, retaining binding affinity and therapeutic efficacy. The flexibility of the described approach makes it easy to vary the nature of the probes for other combinations of diagnostic and therapeutic compounds.

## 1. Introduction

The variable domains of heavy chain only antibodies (VHHs or nanobodies^®^) were discovered in the early 1990s. They are an interesting class of molecules for tumor imaging and therapy, mainly because of their small size (15 kDa versus 150 kDa for monoclonal antibodies) and fast clearance, leading to efficient tissue penetration [1] and good tumor-to-background ratios at early time points post injection [2,3,4,5,6,7,8,9,10]. Furthermore, they are easily expressed recombinantly in bacterial or yeast systems, and very stable under harsh temperature or pH conditions.

To functionalize VHHs with diagnostics or therapeutic groups of interest, various conjugation methods have been employed. The most commonly used method is conjugation to naturally occurring lysine residues. Since often multiple lysines are present in VHHs, there is no control over the stoichiometry. Furthermore, if lysine residues are present in the antigen binding domain, conjugation can interfere with target binding. To this end, unpaired cysteines have been introduced at the C-terminus of the VHH, which can be site-specifically conjugated through maleimide chemistry [11]. Importantly, site-specific labeling was shown to lead to preferable pharmacokinetics and tumor targeting potential [12].

A second site-specific conjugation method that is increasingly employed is the introduction of imaging probes or orthogonal clickable groups through sortase A-mediated transpeptidation of the LPETG amino acid sequence introduced in proteins [13,14,15,16]. The clickable groups enable further modification by cycloaddition reactions with the orthogonal click partner (e.g., another VHH, a fluorophore or a therapeutic molecule).

Since the value of a theranostic approach, in which a targeting molecule can be used for both diagnosis and therapy, is increasingly appreciated, it would be advantageous to conjugate two functional groups to a VHH. Examples are a chelator for labeling with a radiometal and a fluorophore for either fluorescence guided surgery or targeted photodynamic therapy (tPDT). The radiometal can be used for nuclear imaging to localize and quantify target expression preoperatively, and for radioguidance to the tumor intraoperatively. Fluorescence imaging and tPDT can be used during surgery as well, to ensure complete resection and eradication of all tumor cells, respectively [17,18,19,20]. Besides application in the clinic, the presence of a radiometal is also advantageous for performing relatively easy and quantitative biodistribution studies of small compounds in vivo.

To achieve dual-labeling, maleimide chemistry and sortase A-mediated transpeptidation can be combined for introduction of two functional groups [21]. This procedure, however, is laborious and requires numerous incubation and purification steps, thereby also complicating clinical translation. Previously, our group has described dichlorotetrazine as a platform for site-specific dual-labeling of proteins [22,23]. We have shown that bimodal tetrazine-based probes, containing a chelator and a fluorophore, can readily be conjugated to a protein carrying a reactive alkene or alkyne group through the inverse electron demand Diels-Alder (iedDA) reaction. This method was illustrated by preparation of a bimodal anti-HER2 monoclonal antibody for single photon emission computed tomography (SPECT) and fluorescence imaging [22].

In the current study, we performed the experiments with the extensively validated EGFR-targeting VHH 7D12 as a model. 7D12 has been investigated preclinically for cancer treatment as it sterically hinders natural ligand EGF from binding to EGFR [24], thereby inhibiting receptor activation and reducing cell proliferation [25,26]. Furthermore, 7D12 has been used for tumor imaging [7,27,28] and tPDT [29,30,31,32]. Here, we employed the dichlorotetrazine platform for site-specific conjugation of the chelator diethylenetriaminepentaacetic acid (DTPA) and the photosensitizer IRDye700DX to 7D12 for application in a bimodal approach. We evaluated the function of the VHH for SPECT imaging and tPDT of EGFR-expressing tumors.

## 2. Results

### 2.1. Site-Specific Functionalization of Anti-EGFR VHH 7D12 with DTPA and IRDye700DX

The bimodal probe DTPA-NH-Tz-S-IRDye700DX was synthesized in four steps (Appendix A). The detailed synthesis of the precursors as well as their analytical characterization are provided as supporting information (pages 6–11, Appendix A). In short, DTPA(tBu)_4_ ester was condensed with ethylenediamine to obtain DTPA-NH_2_ after acidic deprotection (Appendix A). Then dichlorotetrazine was substituted with DTPA-NH_2_ at room temperature (Appendix A), and with Boc-mercaptoethylamine at 75 °C (Appendix A). Finally, IRDye700DX-NHS was coupled to the free amine group of the mercaptoethylamine at 25 °C to obtain DTPA-NH-Tz-S-IRDye700DX with an overall yield of 18% (Appendix A).

The conjugate DTPA-IRDye700DX-7D12 was obtained in two steps (Figure 1). Detailed description and analysis is described in the supporting information (pages 12–16, Appendix A). First, BCN-PEG_2_-maleimide (Appendix A) was site-specifically coupled to the unpaired cysteine residue of 7D12 (Appendix A). The conversion was complete with a protein recovery of 74% and a degree of labeling (DOL) of 1. The subsequent click reaction between the bicyclonyne (BCN) and the tetrazine group of DTPA-NH-Tz-S-IRDye700DX yielded the bimodal conjugate with a protein recovery yield of 43% and DOL of 0.56 (Appendix A).

Spectral properties of IRDye700DX were retained after conjugation to 7D12 (Figure 2 and Appendix A), with excitation and emission maxima of 689 nm and 698 nm, respectively. The fluorescence quantum yields of DTPA-IRDye700DX-7D12, DTPA-NH-Tz-S-IRDye700DX and IRDye700DX alone, were 0.12, 0.21 and 0.29, respectively. DTPA-IRDye700DX-7D12 produced singlet oxygen upon illumination with 690 nm light, albeit significantly lower when compared to non-conjugated IRDye700DX (*p* < 0.001, Figure 2B).

### 2.2. [^111^In]In-DTPA-IRDye700DX-7D12 Binds Specifically to EGFR-Expressing Cells

For in vitro assays, DTPA-IRDye700DX-7D12 was labeled with 0.25 MBq/µg ^111^In, yielding >98% radiochemical purity. Labeling was stable in PBS and human serum in presence or absence of a molar excess of EDTA at 37 °C for up to 24 h (Figure 3). Furthermore, LCMS performed after incubating the conjugate at radiolabeling-like conditions indicated stability of the product (Appendix A).

[^111^In]In-DTPA-IRDye700DX-7D12 bound to A431 cells, with 11.20 ± 0.33%, 11.36 ± 1.95% and 7.49 ± 0.54% of added activity being membrane bound after 1, 4 and 16 h, respectively (Figure 4A). The tracer was efficiently internalized, with 7.85 ± 0.81%, 14.68 ± 0.88% and 16.14 ± 1.09% of added activity being intracellular after 1, 4 and 16 h, respectively. The minimal binding and internalization in presence of an excess of unlabeled 7D12 illustrated EGFR-specificity. Half maximal inhibitory concentrations (IC_50_) of native 7D12 and DTPA-IRDye700DX-7D12 were 22.76 nM and 18.11 nM, respectively (Figure 4B).

### 2.3. DTPA-IRDye700DX-7D12 Induces Cell Death upon Illumination

After incubation with DTPA-IRDye700DX-7D12, A431 cells were effectively killed upon illumination with 690 nm light. The effect was dependent on protein dose and total light dose in J/cm^2^, whereas no clear correlation for light dose rate and efficacy was found (Figure 5A). To investigate the contribution of membrane bound and internalized fractions of the conjugate to cytotoxicity, cells that contain both fractions, and cells that only contain the internalized fraction were illuminated (Figure 5B). Cell viability was dose-dependently decreased by the intracellular fraction, and this effect was significantly larger when adding the membrane bound fraction (*p* < 0.001 for 1.66, 5 and 15 nm when comparing intracellular and membrane bound with intracellular only), indicating that both fractions contribute to cytotoxicity. Singlet oxygen scavenger *N*-acetylcysteine (NAC) significantly reduced efficacy of the treatment after incubation with up to 1.66 nM of the conjugate (*p* < 0.001) (Figure 5B). Only scavenging of extracellular singlet oxygen contributed to this effect, as no change in cytotoxicity was observed when using cells containing only the internalized fraction of the conjugate (Figure 5B). Finally, cells were incubated with either [^111^In]In-DTPA-IRDye700DX-7D12 or unlabeled DTPA-IRDye700DX-7D12. Both constructs induced cell death with equal efficiency upon illumination. For both constructs cytotoxic effect was absent without illumination (Figure 5C).

### 2.4. [^111^In]In-DTPA-IRDye700DX-7D12 Homes to EGFR-Expressing Tumors and Can Be Visualized with Imaging

A protein dose escalation was performed in mice carrying subcutaneous A431 xenografts. Injection of 1, 5, 25 or 50 µg [^111^In]In-DTPA-IRDye700DX-7D12 (1 MBq) resulted in comparable relative tumor uptake of 4.76 ± 0.3, 4.5 ± 1.16, 5.16 ± 0.48 and 4.54 ± 0.64%IA/g, respectively (Figure 6A). Uptake was partly EGFR specific, as demonstrated by significantly lower tumor uptake upon co-injection of unlabeled cetuximab (4.76 ± 0.3%IA/g versus 2.94 ± 0.79%IA/g, *p* = 0.0201). Furthermore, high uptake in the kidneys, and uptake in the liver and spleen was observed for all concentrations, which could not be blocked by unlabeled cetuximab. The injected protein dose of 50 µg led to the highest absolute tumor uptake (2.27 ± 0.31 µg/g) (Figure 6B). Micro-SPECT (Figure 6C) and near infrared fluorescence imaging (Figure 6D) w ere performed, and the tracer could clearly be visualized with both modalities.

## 3. Discussion

It is well established that protein labeling can significantly alter the targeting properties and pharmacokinetics of proteins, especially in case of small proteins such as VHHs [12]. Here, we have demonstrated that the dichlorotetrazine platform can be used to introduce two modalities, a chelator and the fluorophore IRDye700DX, at a single site of a VHH.

The cysteine residue introduced at the C-terminus of anti-EGFR VHH 7D12 was equipped with BCN-PEG_2_-maleimide to ensure selective labeling with the bimodal tetrazine probe. In future experiments, substitution of the BCN with the more reactive group transcyclooctene (TCO) could increase labeling efficiency [33]. Though IRDye700DX is a photostable phthalocyanine with favorable spectral properties and excellent water solubility [34], it is also a fragile and expensive molecule, and therefore we have enabled addition of IRDye700DX during the last stage of synthesis. The spectral properties of IRDye700DX were preserved after conjugation to 7D12, and conjugation did not affect 7D12 binding affinity as illustrated by similar IC_50_ values. The production of singlet oxygen upon illumination decreased upon conjugation but remained adequate for application in PDT.

Other efforts in site-specific dual-labeling entail sortase A-mediated labeling combined with maleimide [21] or π-clamp conjugation [35], however these procedures include multiple conjugation and purification steps, and in each step material is lost. Furthermore, our current approach ensures a 1:1 ratio between the radiolabel and the PDT dye conjugated to the protein, which guarantees a match between the SPECT signal and the area treated by PDT.

The scope of this study was to develop and implement a technique for site-specific dual-labeling of small proteins such as VHHs. For actual application of bimodal VHHs in a theranostic approach as described here, there are various factors to take into account. First, the rapid renal clearance of VHHs ensures since high tumor-to-background ratios at early time points post injection, but can also lead to low absolute tumor uptake, which negatively affects therapeutic efficacy. Alterations to elongate blood half-life [36], such as conjugation to polyethylene glycol (PEG) or albumin binding domains, are interesting in this respect. Furthermore, injection of high protein doses to achieve high absolute tumor uptake is convenient for therapeutic approaches, however it can induce receptor saturation and thereby decrease signal obtained in nuclear imaging. Finally, an optimal compromise between various factors such as protein dose, size, circulation time and the resulting absolute tumor uptake has to be found.

Ultimately, the bimodal VHH as described here can be applied in image guided therapy, by guiding the surgeon to a tumor lesion by radioactive signal, and by using photosensitizer’s fluorescence and therapeutic effects for ablation of tumor cells. This therapy could be applied in oncologic surgical interventions in EGFR-expressing tumors, such as head-and-neck cancer, bladder cancer, lung cancer and brain tumors. Conjugates of EGFR-targeting monoclonal antibodies with IRDye700DX are currently being evaluated in a clinical setting for treatment of head-and-neck squamous cell carcinoma [37,38,39]. Due to their small size, VHHs have the advantage of more homogenous tissue penetration, leading to more efficient tumor targeting, also when applied in PDT [1,29]. Furthermore, binding to only a small epitope of EGFR, which completely overlaps with the EGF binding site, and the directly induced cytotoxic effect could prevent resistance through mutations of the EGFR ectodomain, which are frequently induced by therapeutic antibodies [40]. Though PDT has been shown to help overcoming drug resistance by for example stimulation of anti-tumor immunity and enhancement of drug delivery, intrinsic and acquired resistance mechanisms to PDT have been described as well [41]. These photobiological aspects of EGFR-directed PDT warrant more thorough investigation.

## 4. Materials and Methods

### 4.1. Production and Characterization of the Anti-EGFR VHH 7D12

Anti-EGFR VHH 7D12 with a C-terminal CLPETG tag for maleimide and sortase A conjugations, a hexa-histidine for purification and a VSV tag for antibody-based detection was expressed and purified as described previously [30,42]. In short, ER2566 bacteria (stock was kindly provided by dr. W.P.J. Leenders) were grown in 2xTY medium, and VHH expression was induced with 1 mM IPTG for 4 h at 30 °C. Subsequently the periplasmic fraction was collected, and VHHs were purified using Ni-NTA sepharose (IBA, Göttingen, Germany). Purity and size were characterized with sodium dodecyl sulfate polyacrylamide gel electrophoresis (SDS-PAGE) under reducing conditions and a coomassie brilliant blue staining. The concentration of the 7D12 in g/L was calculated by UV spectrophotometry by using the extinction coefficient of 7D12 at 280 nm (35535 L·mol^−1^·cm^−1^).

### 4.2. Synthesis of the Bimodal Probe

The synthesis of the bimodal probe DTPA-NH-Tz-S-IRDye700DX was carried out using the methods described previously [22]. The detailed synthesis of the precursors as well as their analytical characterization are provided as supporting information (pages 6–11).

### 4.3. Synthesis of 7D12-BCN

A detailed description of the synthesis of 7D12-BCN as well as the analytical characterization of the compound is provided in the supporting information (pages 12–14). To a 7.24 g/L solution of VHH 7D12 in PBS (2 mg, 110 nmol) a 20 mM solution of tris(2-carboxyethyl)phosphine (TCEP) in PBS (637 μg, 2.24 μmol, 20 equiv.) and a 20 mM solution of BCN-PEG_2_-maleimide in DMSO (1.06 mg, 2.24 μmol, 20 equiv.) were added. The solution was stirred at 37 °C in a thermomixer (900 rpm) for 7 h. An additional amount of TCEP (318.5 μg, 1.12 μmol, 10 equiv.) and BCN-PEG_2_-maleimide in DMSO (795 μg, 1.68 μmol, 15 equiv.) were added and the solution was stirred (900 rpm, 37 °C) overnight. Excess of TCEP and BCN was removed by ultrafiltration on an Ultra Ultracel 3 kDa (Amicon, Merck Millipore, Burlington, MA, USA). This ultrafiltration step also allowed the product to be concentrated and the DMSO to be removed. The VHH 7D12-BCN was obtained as a 3.69 g/L solution in PBS (1.48 mg, protein recovery = 74%). The degree of labeling was determined to be 1 by MALDI-TOF and HPLC-MS analyses.

### 4.4. Site-Specific Labeling of 7D12-BCN with DTPA and IRDye700DX

To a 3.69 g/L solution of VHH 7D12-BCN in PBS (922 μg, 50.4 nmol), 75.2 μL of a 6.7 mM stock solution of DTPA-NH-Tz-S-IRDye700DX in PBS (1.18 mg, 504 nmol, 10 equiv.) and 905 μL of PBS were added. The solution was stirred in a thermomixer (900 rpm, 37 °C) overnight. The product was purified by FPLC (Äkta Pure 25 M chromatography system, GE Healthcare Life Sciences, Chicago, IL, USA) on a Superdex 75 10/300 GL column (cross-linked agarose and dextran, 13 μm, 10 × 300–10 mm, 24 mL), at 0.8 mL/min, with phosphate buffer (20 mM, pH 7.3) with NaN_3_ (0.05%) as eluent (Appendix A). The bimodal conjugate DTPA-IRDye700DX-7D12 was obtained as a 1.27 g/L solution (444 μg, protein recovery = 43%). The degree of labeling was determined to be 0.56 (UV-visible spectroscopy). A correction factor (0.1) was introduced to account for the contribution of the phthalocyanine absorbance at 280 nm. The molar concentration of the fluorophore was calculated by UV spectrophotometry by using the extinction coefficient of IRDye700DX at 689 nm in PBS (165000 L·mol^−1^·cm^−1^). The analytical characterization of the compound is provided in the supporting information (Appendix A).

### 4.5. Characterization of the Conjugate

Spectral characterization of the compounds is provided in the supporting information (Appendix A).

UV-visible spectra were recorded on a Cary 60 scan (double beam) spectrophotometer (Varian, Palo Alto, CA, USA) by using a rectangular quartz cell (100-QS, 45 × 12.5 × 12.5 mm, pathlength: 10 mm, chamber volume: 3.5 mL, Hellma-France, Paris, France), at 25 °C (using a temperature control system combined with water circulation). Fluorescence spectroscopic studies were performed with a Fluorolog spectrofluorometer (HORIBA Jobin Yvon, Palaiseau, France, software FluorEssence) at 25 °C, with a standard fluorometer cell (LB Q, light path: 10 mm, width: 10 mm, chamber volume: 50 μL, Labbox, Barcelona, Spain).

UV-Visible absorbance and fluorescence spectra of the IRDye700DX dye and related conjugates were recorded in the range 250–800 nm in PBS (pH 7.4, 0.01 M) with concentrations in the micromolar range. Emission spectra were recorded in the range 630–850 nm after excitation at 620 nm (shutter: Auto Open, Ex. Slit = 5 nm and Em. slit = 5 nm). Excitation spectra were recorded in the range 250–760 nm with emission measurement at 770 nm (shutter: Auto Open, Ex. slit = 5 nm and Em. slit = 5 nm). Fluorescence quantum yields were measured at 25 °C by a relative method using a suitable standard (AzaBODIPY JPL04, Φ = 0.36, dilution by a factor of 3 between absorption and fluorescence measurements). The following equation was used:

ΦF_x_ = (A_S_/A_X_)(F_X_/F_S_)(n_X_/n_S_)^2^ΦF_s_ where A is the absorbance (in the range of 0.01–0.1 A.U.), where F is the area under the emission curve, n is the refractive index of the solvents (at 25 °C) used in measurements, and the subscripts s and x represent standard and unknown, respectively.

### 4.6. Radiolabeling of DTPA-IRDye700DX-7D12 and Quality Control

DTPA-IRDye700DX-7D12 was incubated with [^111^In]InCl_3_ (Curium, Petten, The Netherlands) and twice the volume of 0.5 M 2-(N-morpholino)ethanesulfonic (MES) buffer, pH 5.5 for 30 min at room temperature. Labeling efficiency and radiochemical purity were determined by instant thin-layer chromatography (iTLC) on a silica gel chromatography strip (Biodex, Shirley, NY, USA), using 0.1 M citrate buffer pH 6.0 as the mobile phase. For stability and in vitro studies, labeling was performed at 0.25 MBq/µg. Stability of labeling up to 24 h was determined by iTLC after incubating 180 nM of the labeled construct in either human serum or PBS with or without a thousand-fold molar excess of ethylenediaminetetraacetic acid (EDTA). Furthermore, stability of the construct in labeling conditions was determined by HPLC-MS analyses after 0, 2 and 4 h (supporting information, Appendix A). For in vivo studies, labeling was performed at 3 MBq/µg and the labeled product was purified on a PD-10 column (GE, Woerden, The Netherlands), that was eluted with PBS 0.5% BSA.

### 4.7. Singlet Oxygen Production by DTPA-IRDye700DX-7D12 upon Illumination

DTPA-IRDye700DX-7D12 (250 nM) was incubated with p-nitrosodimethylaniline (RNO; 50 µM) and imidazole (400 µM) as an acceptor of ^1^O_2_ in phosphate buffered saline pH 7.4 in clear flat-bottom 96-wells plates (Corning Costar, Corning, NY, USA), and illuminated with 690 nm light at 200 mW/cm^2^. Absorbance at 440 nm was measured every minute with the Tecan Infinite^®^ 200 Pro (Tecan, Männedorf, Switzerland), to determine ^1^O_2_ induced bleaching of RNO.

### 4.8. Cell Culture

The human epidermoid carcinoma cell line A431, carrying an amplification of the EGFR gene, [43] was cultured in RPMI-1640 (GIBCO, Thermo-Fisher Scientific, Waltham, MA, USA), supplemented with 2 mmol/L L-glutamine (GIBCO, Thermo-Fisher Scientific) and 10% FCS (Sigma-Aldrich, St. Louis, MO, USA). During all in vitro assays, cells were incubated at 37 °C, in a humidified atmosphere with 5% CO_2_, unless stated otherwise.

### 4.9. In Vitro Binding and Internalization of DTPA-IRDye700DX-7D12

A431 cells were plated at 60,000 cells/well in 6-wells plates and grown to 80% confluency in 3 days. Cells were incubated with 1600 Bq [^111^In]In-DTPA-IRDye700DX-7D12 in RPMI with 0.5% BSA (RPMI-BSA) for 1, 4 and 16 h. EGFR specificity of binding was assessed by co-incubation with 15 µg unlabeled 7D12 per well. After incubation, cells were washed twice with PBS, and the receptor-bound [^111^In]In-DTPA-IRDye700DX-7D12 was retrieved by incubation with ice-cold 0.1 M acetic acid, 154 M NaCl, pH 2.6 for 10 min on ice. After washing twice with PBS, cells containing the internalized [^111^In]In-DTPA-IRDye700DX-7D12 were collected with 1 mL 0.1 M NaOH. Activity in both fractions was counted in a γ-counter (2480 Wizard 3″, LKB/Wallace, Perkin-Elmer, Boston, MA, USA). Specific binding and internalization were calculated by subtracting the non-specific signal from the total signal.

### 4.10. IC_50_ Assay

A431 cells were plated at 60,000 cells/well in 6-wells plates, and grown to 80% confluency in 3 days. Cells were incubated with increasing concentrations of either unlabeled DTPA-IRDye700DX-7D12 or native 7D12 (0.005 to 450 nM) in RPMI-BSA, in presence of 1600 Bq of [^111^In]In-DTPA-IRDye700DX-7D12 for 4 h on ice. Subsequently, cells were washed twice with PBS, and collected with 1 mL 0.1 M NaOH. Activity in all fractions was counted in a γ-counter (2480 Wizard 3″, LKB/Wallace, Perkin-Elmer). The IC_50_ values (the concentrations of DTPA-IRDye700DX-7D12 or 7D12 needed to replace 50% of the bound ^111^In-labeled DTPA-IRDye700DX-7D12) were calculated in GraphPad Prism version 5.0 (GraphPad Software, San Diego, CA, USA).

### 4.11. In Vitro Photodynamic Therapy—^111^In-Labeled Versus Unlabeled DTPA-IRDye700DX-7D12

A431 cells were plated at 5000 cells per well in 96 wells plates and grown to 50% confluency in 2 days. Then cells were incubated for 4 h with increasing concentrations of ^111^In-labeled or unlabeled DTPA-IRDye700DX-7D12 in RPMI-BSA. Upon washing twice with PBS, culture medium was added to the wells, and cells were illuminated with 90 J/cm^2^ 690 nm light at a dose rate of 200 mW/cm^2^ using a light emitting diode. Non-illuminated cells were taken as control to determine dark toxicity of the conjugates. The next day, cell viability was determined with the CellTiter-Glo^®^ luminescence cell viability assay (Promega, Madison, WI, USA), according to manufacturer’s protocol.

### 4.12. In Vitro Photodynamic Therapy—The Effect of Acid Wash, Light Dose (Rate) and Singlet Oxygen Scavenging

A431 cells were plated at 5000 cells per well in 96 wells plates, and grown to 50% confluency in 2 days. Then cells were incubated for 3 h with increasing concentrations of unlabeled DTPA-IRDye700DX-7D12 in RPMI-BSA. Upon washing twice with PBS, part of the wells were incubated with ice-cold 0.1 M acetic acid, 154 M NaCl, pH 2.6 for 10 min on ice to remove the membrane bound fraction of the conjugate. Upon washing twice with PBS, RPMI-BSA with or without 15 mM *N*-acetyl cysteine (NAC) (Thermo-Fisher Scientific, Waltham, MA, USA) was added to the wells and incubated for 1 h. Subsequently, cells were illuminated with various light doses (30 J/cm^2^–90 J/cm^2^) at various light dose rates (50 mW/cm^2^–200 mW/cm^2^) using a 690 nm light-emitting diode [44]. RPMI-BSA was replaced with culture medium. The next day, cell viability was determined with the CellTiter-Glo^®^ luminescence cell viability assay (Promega, Madison, WI, USA), according to manufacturer’s protocol.

### 4.13. Ex Vivo Biodistribution and In Vivo SPECT and Fluorescence Imaging

Animal studies were approved by the Central Authority for Scientific Procedures on Animals (RU-DEC-2015-0071) and carried out under supervision of the local Animal Welfare Body. Tumor targeting by DTPA-IRDye700DX-7D12 was determined in a biodistribution and imaging study in mice bearing A431 xenografts. Mice (n = 3 per group) were injected subcutaneously on the right shoulder with 2.5 × 10^6^ A431 cells in 200 µL RPMI. After 9 days of tumor growth, when tumor size was ~100–200 mm^3^, mice were injected intravenously with various protein doses of [^111^In]In-DTPA-IRDye700DX-7D12 (1 MBq in PBS 0.5% BSA). To determine the specificity of EGFR targeting, a separate group (n = 3) was injected intravenously with a blocking dose of 1 mg unlabeled cetuximab at 48 h prior to 1 µg [^111^In]In-DTPA-IRDye700DX-7D12 injection. Mice were euthanized by CO_2_ suffocation at 16 h after tracer injection, and near infrared fluorescence imaging of the mice was performed using the IVIS Lumina closed-cabinet fluorescence scanner (Caliper LifeSciences, Waltham, MA USA), excitation 640 nm, autofluorescence correction excitation 535 nm; both measured with the Cy5.5 filter). Tumors and other tissue samples (blood, muscle, liver, lung, kidney, spleen, pancreas and stomach) were harvested and weighed, and radioactivity in these samples was determined in a γ-counter (2480 Wizard 3″, LKB/Wallace, Perkin-Elmer, Waltham, MA, USA). Radioactivity concentrations were calculated as percentage of the injected activity per gram of tissue (%IA/g) and corrected for decay using injection standards. Additionally, one mouse was injected with 3 µg DTPA-IRDye700DX-7D12 labeled with 10 MBq indium-111. 19 h post injection it was euthanized and imaged for 1 h using a U-SPECT/CT-II (MILabs, Utrecht, The Netherlands). The image was acquired using a 1 mm diameter pinhole ultra-high sensitivity mouse collimator. The SPECT scan was reconstructed using software from MILabs, using an 0.4 mm voxel size, one iteration and 16 subsets, and visualized in Inveon Research Workplace software (version 3.0; Siemens Preclinical Solutions, München, Germany).

### 4.14. Statistics

Graphpad Prism was used for statistical analyses. Statistical significance was determined with unpaired *t*-test or two-way ANOVA with post-hoc Bonferroni. * = *p* < 0.05; ** = *p* < 0.01 *** = *p* < 0.001.

## 5. Conclusions

In conclusion, we have applied an innovative method for site-specific dual-labeling of an unpaired cysteine in VHHs. This approach can easily be translated to other small proteins that are interesting for imaging and therapy such as DARPins and affibodies. Furthermore, as it is easy to vary the nature of the bimodal probe, it is a highly flexible system for introduction of a variety of therapeutic or diagnostic modalities, without affecting the target-binding domain of the proteins. Application of this conjugation method increases the feasibility of using bimodal small proteins in the field of tumor theranostics.

## Figures and Tables

**Figure 1 cancers-13-00428-f001:**
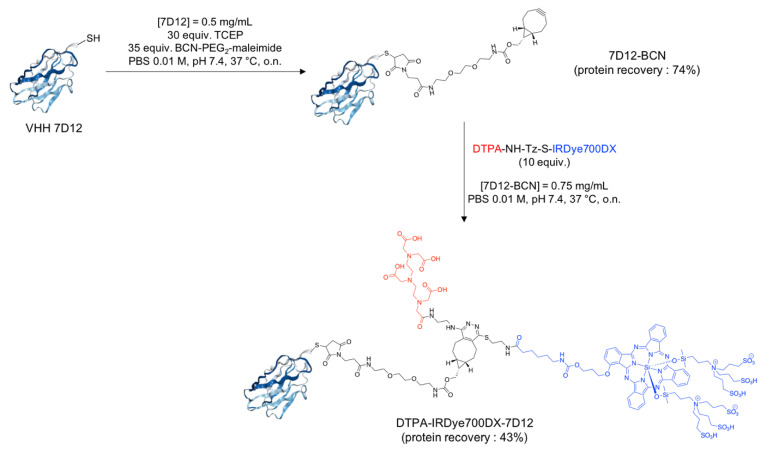
Synthesis of the bimodal conjugate DTPA-IRDye700DX-7D12.

**Figure 2 cancers-13-00428-f002:**
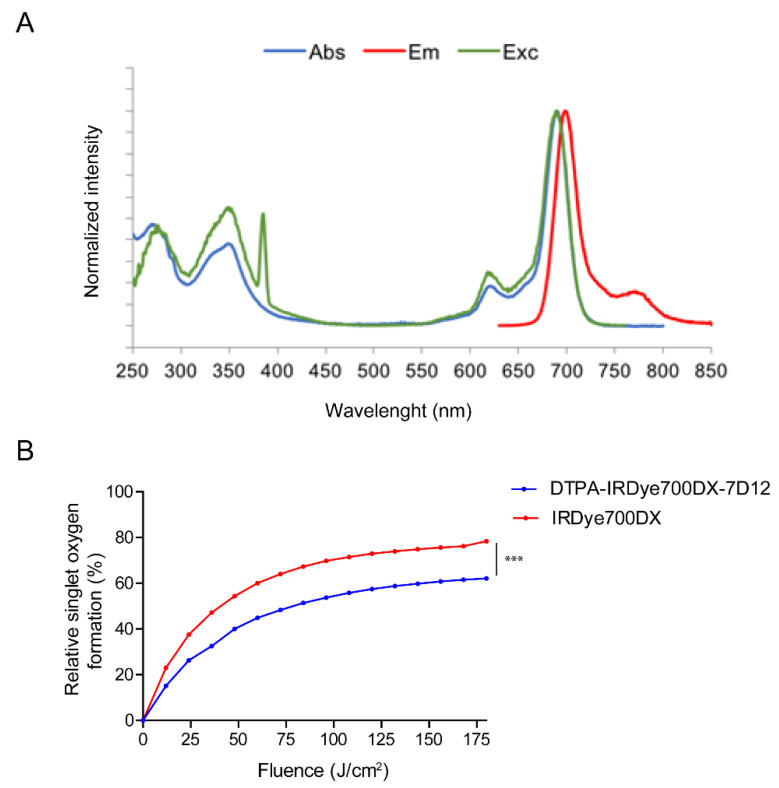
(**A**) Normalized photophysical spectra (absorbance, excitation and emission) of bimodal conjugate DTPA-IRDye700DX-7D12. (**B**) Formation of singlet oxygen upon illumination with 200 mW/cm^2^ 690 nm light by IRDye700DX and conjugate DTPA-IRDye700DX-7D12. *** *p* < 0.001, as determined with two-way ANOVA for repeated measures with Bonferroni post-hoc test.

**Figure 3 cancers-13-00428-f003:**
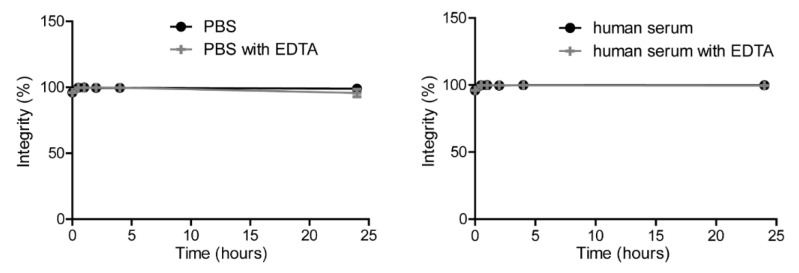
Stability of [^111^In]In-DTPA-IRDye700DX-7D12 in PBS (left graph) or human serum (right graph) in presence or absence of 1000-fold molar excess EDTA.

**Figure 4 cancers-13-00428-f004:**
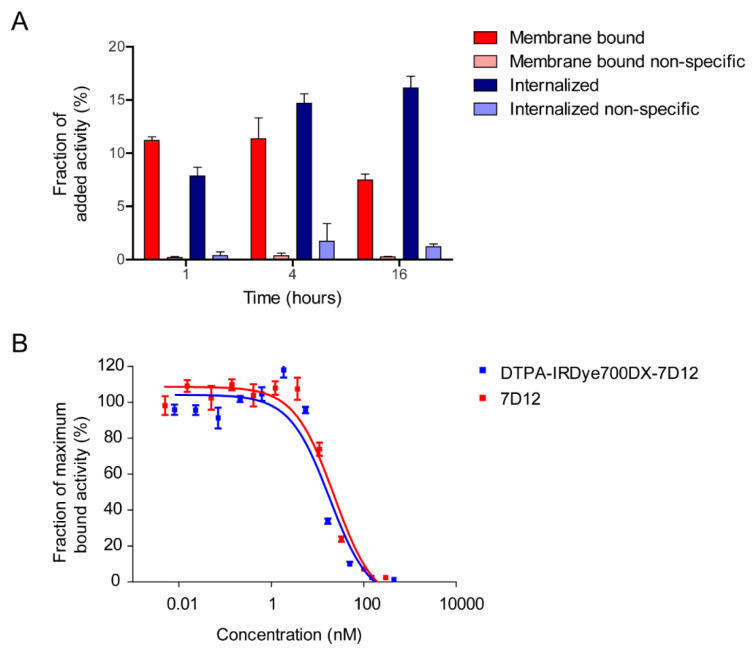
(**A**) Bound and internalized fractions of [^111^In]In-DTPA-IRDye700DX-7D12 after 1, 4 and 16 h of incubation at 37 °C, in absence or presence of an excess of unlabeled native 7D12. (**B**) Competition of cell associated [^111^In]In-DTPA-IRDye700DX-7D12 by increasing concentrations of native 7D12 or unlabeled DTPA-IRDye700DX-7D12.

**Figure 5 cancers-13-00428-f005:**
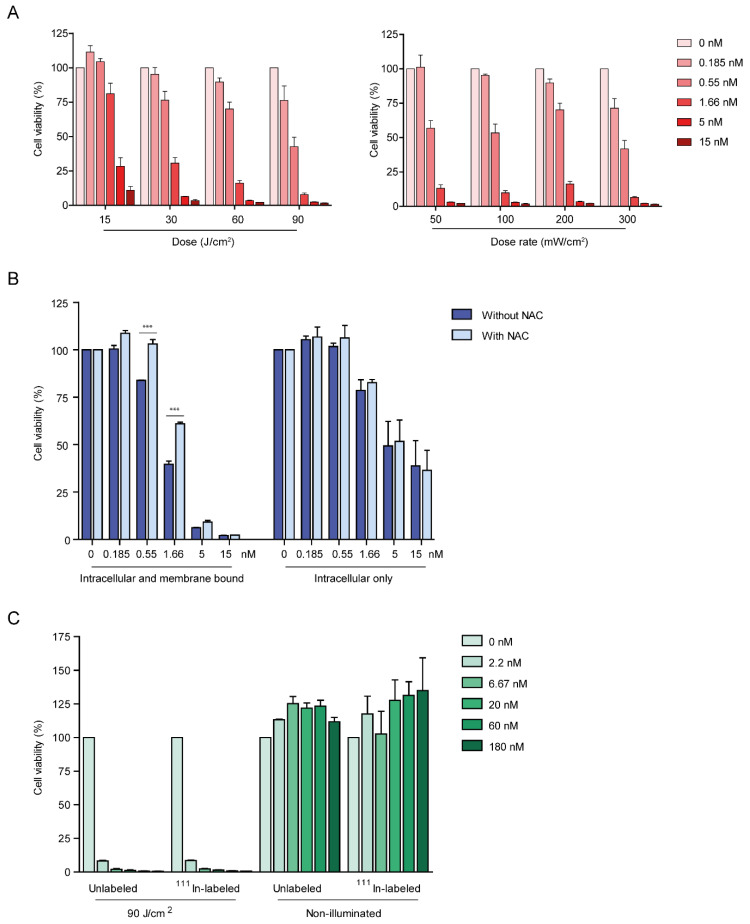
(**A**) DTPA-IRDye700DX-7D12 induces cell death of A431 cells upon illumination with varying light doses at a fixed light dose rate of 200 mW/cm^2^ (left) and varying light dose rates (right) at a fixed dose of 60 J/cm^2^ of 690 nm light. (**B**) Both internalized as well as membrane bound fractions of the conjugate contribute to light induced toxicity upon illumination with 60 J/cm^2^ 690 nm light at 200 mW/cm^2^. Incubation with singlet oxygen scavenger NAC partly inhibits the cytotoxic effect in cells containing both fractions, but not in cells containing only the intracellular fraction. (**C**) Both [^111^In]In-DTPA-IRDye700DX-7D12 and unlabeled DTPA-IRDye700DX-7D12 induce toxicity with equal efficiency upon illumination with 90 J/cm^2^ 690 light at 200 mW/cm^2^, without illumination no toxicity is observed. Significance is determined with two-way ANOVA with post-hoc Bonferroni *** = *p* < 0.001.

**Figure 6 cancers-13-00428-f006:**
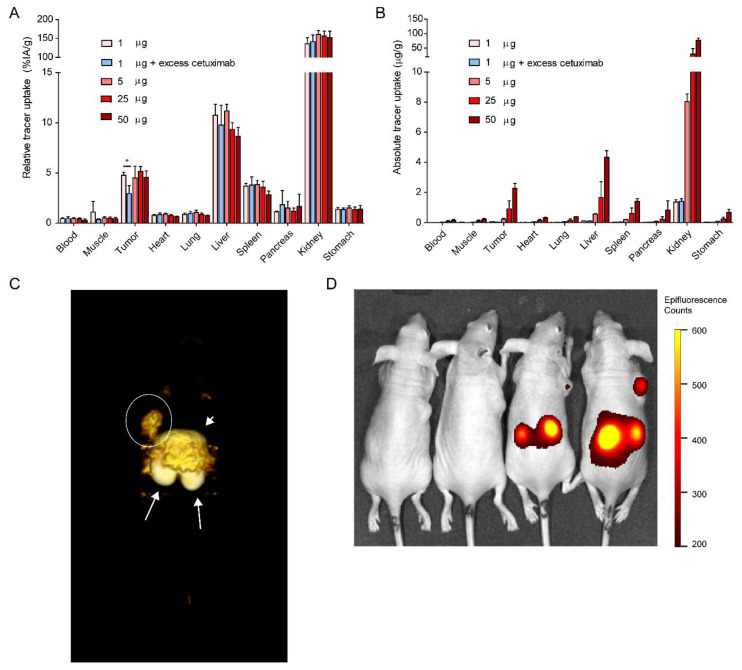
Ex vivo biodistribution of various protein doses of [^111^In]In-DTPA-IRDye700DX-7D12 (n = 3 per group) showing (**A**) relative tracer uptake and (**B**) absolute tracer uptake in EGFR-expressing tumors and other tissues. Statistical significance is determined with an unpaired *T*-test, * = *p* < 0.05. (**C**) µSPECT of a mouse 19 h after injection of 3 µg 10 MBq [^111^In]In-DTPA-IRDye700DX-7D12. Note the high uptake in kidneys (arrows), liver (arrowhead) and tumor (circle). (**D**) Near infrared fluorescence image of mice 16 h after injection of 1, 5, 25 or 50 µg 1 MBq [^111^In]In-DTPA-IRDye700DX-7D12. At higher protein concentrations, uptake in kidneys and tumor is visualized.

## Data Availability

The data presented in this study are available on request from the corresponding author, as no public datasets for these preclinical data are generated.

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
