# Peer review of "Site-Specific Dual-Labeling of a VHH with a Chelator and a Photosensitizer for Nuclear Imaging and Targeted Photodynamic Therapy of EGFR-Positive Tumors"

_cancers, 2021, doi:10.3390/cancers13030428_

Round 1

Reviewer 1 Report

The article describes a theranostic application of anti-EGFR VHH antibodies, potentially useful in the treatment/diagnostic of cancer. The authors have developed in the study presented a method for site-specific dual-labelling of a VHH cysteine, which appears more efficient compared with already existing procedures.
The study is overall well presented, and experiments support conclusions.
It is suggested a revision of the spelling of words, as hyphens' inconsistent use was observed throughout the main text and the supplementary material.
Supplementary material should be reorganized and improved. It is particularly suggested to label more clearly every supplementary figure, including proper numbering and extensive figure legends, and refer in the main text to supplementary experimental material by using the figures' numbers, instead than pages. In the supplementary should also be included the Äkta FPLC purification chromatogram of the product.

Reviewer 2 Report

The manuscript entitled “Site-specific dual-labeling of a VHH with a chelator and a photosensitizer for nuclear imaging and targeted photodynamic therapy of EGFR-positive tumors” by Renard et al, aimed to explore the employment of the dichlorotetrazine platforma for the site-specific dual conjugation to epidermal growth factor receptor (EGFR) VHH 7D12.

The manuscript is professionally written, and the introduction extensively deals with the main strategies known for the site-specific conjugation of VHH. Their chemical method for site-specific dual-labeling involving unpaired cysteine could be useful for other nanobody-based therapies.

Minor comment:

The introduction must be included also information concerning the mechanism of action of VHH 7D12 (its binding to the extracellular region of EGFR inhibits ligand-induced receptor activation and tumor growth; it is a ligand competitive inhibitor of EGFR. See doi:10.1016/j.str.2013.05.008).

The nanobody 7D12 interacts to a very small epitope that almost exclusively consists of amino acids functionally relevant for EGF binding (DOI: 10.1002/ijc.26145; doi:10.1016/j.str.2013.05.008). Cancer therapies, including those EGFR-based, generally induce a genetic diversification and a resistance phenomenon occurs at both extracellular (DOI: 10.1158/1535-7163.MCT-18-0849) and intracellular domains (doi: 10.3390/ijms21051721). In this context, the authors should be emphasized the potential application of their chemical strategy in this direction.  

Reviewer 3 Report

The authors aimed on preparation of a probe for combination of diagnosis (nuclear imaging) and therapy (PDT). Therefore, a photosensitizer was used and radiolabeling was applied. Specifically, the synthesized probe is EGFR-targeting. Overall, this probe is well synthesized. However, the biological activity of this probe requires more investigation.

1. In Figure 4A, the measurements were done in the 3rd and the 24th hour. The time interval is too big. How to distinguish internalized and internalized non-specific? How would the membrane bound and internalized fraction affect the function of the prob?

2. In Figure 5, the authors added NAC to decrease ROS generation. Is the added NAC at a reasonable concentration? Would NAC interact with the internalized probes?

3. PDT effect was confirmed in cell viability assay, but no other investigation was done in cellular study. Unfortunately, PDT effect was not investigated in animals.
